# Detection of Circulating Tumor Cells Using the Attune NxT

**DOI:** 10.3390/ijms24010021

**Published:** 2022-12-20

**Authors:** Mandy Gruijs, Carolien Zeelen, Tessa Hellingman, Jasper Smit, Frank J. Borm, Geert Kazemier, Chris Dickhoff, Idris Bahce, Joop de Langen, Egbert F. Smit, Koen J. Hartemink, Marjolein van Egmond

**Affiliations:** 1Amsterdam UMC location Vrije Universiteit Amsterdam, Molecular Cell Biology and Immunology, De Boelelaan 1117, Amsterdam 1081 HV, The Netherlands; 2Cancer Center Amsterdam, Cancer Biology and Immunology, Amsterdam 1081 HV, The Netherlands; 3Amsterdam Institute for Infection and Immunity, Cancer Immunology, Amsterdam 1081 HV, The Netherlands; 4Amsterdam UMC location Vrije Universiteit Amsterdam, Surgery, De Boelelaan 1117, Amsterdam 1081 HV, The Netherlands; 5Netherlands Cancer Institute—Antoni van Leeuwenhoek Hospital, Thoracic Oncology, Plesmanlaan 121, Amsterdam 1066 CX, The Netherlands; 6Leiden University Medical Center, Pulmonology, Albinusdreef 2, Leiden 2333 ZB, The Netherlands; 7Amsterdam UMC location Vrije Universiteit Amsterdam, Cardiothoracic Surgery, De Boelelaan 1117, Amsterdam 1081 HV, The Netherlands; 8Amsterdam UMC location Vrije Universiteit Amsterdam, Pulmonology, De Boelelaan 1117, Amsterdam 1081 HV, The Netherlands; 9Netherlands Cancer Institute—Antoni van Leeuwenhoek Hospital, Surgery, Plesmanlaan 121, Amsterdam 1066 CX, The Netherlands

**Keywords:** cancer, metastasis, CTC, CTC detection method, whole blood, flow cytometry

## Abstract

Circulating tumor cells (CTCs) have been detected in many patients with different solid malignancies. It has been reported that presence of CTCs correlates with worse survival in patients with multiple types of cancer. Several techniques have been developed to detect CTCs in liquid biopsies. Currently, the only method for CTC detection that is approved by the Food and Drug Administration is CellSearch. Due to low abundance of CTCs in certain cancer types and in early stages of disease, its clinical application is currently limited to metastatic colorectal cancer, breast cancer and prostate cancer. Therefore, we aimed to develop a new method for the detection of CTCs using the Attune NxT—a flow cytometry-based application that was specifically developed to detect rare events in biological samples without the need for enrichment. When healthy donor blood samples were spiked with variable amounts of different EpCAM+EGFR+ tumor cell lines, recovery yield was on average 75%. The detection range was between 1000 and 10 cells per sample. Cell morphology was confirmed with the Attune CytPix. Analysis of blood samples from metastatic colorectal cancer patients, as well as lung cancer patients, demonstrated that increased EpCAM+EGFR+ events were detected in more than half of the patient samples. However, most of these cells showed no (tumor) cell-like morphology. Notably, CellSearch analysis of blood samples from a subset of colorectal cancer patients did not detect CTCs either, suggesting that these blood samples were negative for CTCs. Therefore, we anticipate that the Attune NxT is not superior to CellSearch in detection of low amounts of CTCs, although handling and analysis of samples is easier. Moreover, morphological confirmation is essential to distinguish between CTCs and false positive events.

## 1. Introduction

Cancer is one of the main global health problems. The incidence of cancer is approximately 19.3 million patients with a mortality of nearly 10 million patients annually worldwide [1]. Most cancer-associated deaths are due to metastatic disease. During metastasis development, tumor cells disseminate from the primary tumor, invade the surrounding tissue, and enter the circulation. After arrival in a distant organ, tumor cells may grow out as metastasis [2,3,4,5]. Circulating tumor cells (CTCs) have been detected in many patients with different solid malignancies. The presence of CTCs has been associated with both a high primary tumor load and increased risk of metastasis development [6]. It has been demonstrated that CTCs can develop into metastases in murine xenograft models [7,8]. CTCs that were isolated from small cell lung cancer (SCLC) patients were tumorigenic in mice [7]. Similarly, transplantation of CTCs, which were isolated from primary breast cancer patients, in mice induced bone, lung and liver metastases [8].

Over the years, CTCs have gained increasing interest, as they may have prognostic value [9,10]. It has been shown that presence of CTCs was correlated with cancer relapse and worse survival in localized colorectal cancer (CRC) [11]. Meta-analysis demonstrated that presence of CTCs was significantly associated with disease progression and reduced survival. This association has been shown for breast cancer [12,13,14], non-small cell lung cancer (NSCLC) [15], melanoma [16], head and neck cancer [17], bladder cancer [18] and testicular cancer [19]. Presence of CTCs was also strongly associated with worse patient outcome in metastatic CRC [20,21,22]. A meta-analysis indicated a significantly higher incidence of CTCs in patients with metastasis, compared to patients without metastasis [20]. Additionally, patients with CTCs responded significantly less to treatment compared to patients without CTCs. The presence of CTCs was significantly related to poorer survival. This association holds true for metastatic breast cancer [23,24,25,26], prostate cancer [27,28], SCLC [29], NSCLC [30,31] and neuroendocrine tumors [32] as well.

It has been demonstrated that CTCs might be useful in early detection of cancer, as well as in monitoring treatment [9,10]. In a cohort of patients with lung lesions, CTCs were detected in 90% of patients with malignant lesions and in 5% of patients with benign lesions [33]. Therefore, presence of CTCs may potentially distinguish benign lesions from malignant lesions. Similarly, it has been shown that patients with chronic obstructive pulmonary disease, who presented with CTCs during their annual surveillance, later developed lung cancer [34]. The CirCe01 trial assessed the number of CTCs in metastatic breast cancer patients, who started a new line of treatment [35]. Patients with at least 1 CTC per 7.5 mL of blood at baseline were assessed again after their first (before their second) cycle of chemotherapy. A relative decrease of at least 70% in the number of CTCs was associated with enhanced survival. Similarly, it has been demonstrated that serial analysis of CTCs during treatment was a better predictor of survival compared to CTC count at baseline alone [23,26]. Additionally, metastatic CRC patients with high CTC counts were shown to be eligible for treatment intensification, resulting in improved survival [36]. The PRODIGE 17 trial showed that dynamic evaluation of CTCs in patients with advanced esophageal cancer and gastric cancer could provide insight in response to treatment [37]. Similar findings have been observed in patients with metastatic prostate cancer [38]. Moreover, the STIC-CTC trial, which compared a clinician-based choice of first-line therapy in metastatic breast cancer patients with a CTC-based choice, demonstrated increased survival in the CTC-based treatment group [39].

As clinical decisions may be based on the detection of CTCs, it is essential that the used methods are accurate, consistent, reproducible, and reliable. Several techniques have been developed to detect CTCs in liquid biopsies, in particular in blood [40,41]. As genetic alterations are essential in the development of cancerous cells, CTC detection may be based on mutational analysis [42]. It has been demonstrated that (digital) polymerase chain reaction (PCR) can be used in the diagnosis of cancer through identification of cancer-associated mutations in CTCs or circulating tumor DNA (ctDNA). However, this method is only suitable for identification of predefined mutations. Alternatively, aneuploidy—an abnormal amount of chromosomes—is commonly observed in a variety of cancer types [43]. The degree of aneuploidy in cells is proportional to their degree of malignancy, and is inversely correlated with prognosis. It has been demonstrated that detection of aneuploid cells using immunostaining fluorescence in situ hybridization (iFISH) can be used as biomarker for response to treatment and prognosis in NSCLC patients [44]. Increased numbers of aneuploid cells were significantly associated with shortened survival. Other techniques for detection of CTCs are based on either cell morphology, or expression of specific antigens [45,46]. Currently, the only method for CTC detection approved by the Food and Drug Administration (FDA) is CellSearch [47,48]. CellSearch provides a CTC detection platform based on positive enrichment. Red blood cells are removed by density centrifugation, after which cells expressing epithelial cell adhesion molecule (EpCAM) are isolated via magnetic separation. Then, cells are stained with the nuclear stain DAPI, cytokeratin (CK) and CD45. CTCs are defined as DAPI+CK+CD45- cells by microscopy.

Although multiple studies in a wide variety of cancer types have demonstrated CellSearch’ ability to detect CTCs, its clinical application is currently limited to metastatic CRC, breast cancer and prostate cancer. One of the main limiting factors is the low abundance of CTCs in certain cancer types and in early stages of malignant disease. A significant association with reduced survival was found in the presence of ≥5 CTCs per 7.5 mL of blood for metastatic breast cancer and prostate cancer, and ≥3 CTCs per 7.5 mL of blood for metastatic CRC [47]. In the present study, we therefore aimed to develop a new method for the detection of CTCs in patients with CRC and lung cancer. In this method, blood samples are minimally handled and analyzed using the Attune NxT—a flow cytometer particularly suitable for the detection of rare events. This method could yield more CTCs per blood sample compared to CellSearch, while limiting handling time and resources.

## 2. Results

### 2.1. Seventy-Five% of Tumor Cells Can Be Detected Using the Attune NxT

As cancer is a highly heterogeneous disease, patients will present a broad spectrum of CTCs. Additionally, the number of CTCs in a blood sample will differ between patients. Therefore, as a first step, healthy donor blood samples were spiked with decreasing amounts of different tumor cell lines. Then, samples were processed, stained, and analyzed with flow cytometry. When samples were spiked with HCT116 cells, yield was on average 75% (Figure 1). Detection range was between 1000 and 10 EpCAM+EGFR+ events per sample. This was similar when A431 cells, HT29 cells, H1650 cells and HCC827 cells were used (Figure 2). Importantly, yield was similar when independent researchers performed the same experiments as additional control.

To confirm the suitability of EpCAM and EGFR as markers for the detection of (epithelial) CTCs, healthy donor blood samples were spiked with CFSE-labelled HCT116 cells. Then, samples were processed, stained and analyzed with flow cytometry. More than 75% of the CFSE-labeled cells was detected (Appendix A). When samples were stained for the presence of EpCAM+EGFR+ cells, the CFSE+ cell population and the EpCAM+EGFR+ cell population overlapped (Appendix A). This indicates that we can detect the majority of tumor cells in a sample with anti-EpCAM/anti-EGFR antibodies.

Most CTC detection assays, including CellSearch or other flow cytometry-based methods, require sample enrichment. As comparison, healthy donor blood samples were spiked with HT29 cells, and samples were analyzed after EpCAM enrichment. Less than 20% of spiked tumor cells was detected in the eluent of the peripheral blood mononuclear cell (PBMC) fraction, indicating that 80% of the cells was lost (Appendix A). No tumor cells were found in the flowthrough of the PBMC fraction, nor in the plasma fraction, Lymphoprep fraction or polymorphonuclear cell (PMN) fraction (Appendix A). Even when HT29 cells in medium were enriched with EpCAM+ beads, yield was less than 40% (Appendix A), suggesting that 60–80% of tumor cells are lost during EpCAM enrichment, presumably due to retainment in the magnetic column. Thus, EpCAM enrichment might cause underestimation of the number of CTCs. As the Attune NxT does not require sample enrichment, this will likely be less of a problem using the CTC detection method as described here.

### 2.2. Optimization of CTC Detection

As the time between blood collection and processing might differ per sample, it was investigated whether the number of CTCs detected in a sample changed over time. Healthy donor blood was spiked with HCT116 cells and processed either directly or after two hours, four hours or six hours. While the yield in the directly processed sample was over 80% (Figure 3A, first column), this gradually declined to 50%, 30%, and 25% in the samples processed after two hours, four hours and six hours, respectively (Figure 3A, second-fourth column). Moreover, the EpCAM+EGFR+ cell population becomes more scattered over time, supporting that samples should be processed immediately after blood drawl.

Next, we investigated whether the CTC detection rate was similar between blood samples collected in either heparin tubes, EDTA tubes or CPT tubes. A CPT tube contains a gel layer, which separates plasma and PBMCs from erythrocytes and granulocytes upon centrifugation. This system limits both inter-researcher variability and technical variability, as well as processing time. It has been shown that there are no differences in PBMC yield and viability using this method [49]. Healthy donor blood was collected in a heparin tube, an EDTA tube or a CPT tube and spiked with HCT116 cells. Approximately 75% of HCT116 cells was detected in the heparin sample (Figure 3B, left column) compared to 55% in the EDTA sample and the CPT sample (Figure 3B, middle and right column), supporting the use of heparin tubes.

Methods of sample fixation were investigated as well. Healthy donor blood was spiked with HCT116 cells, processed, and fixed with either RBC lysis buffer or 2% PFA for 5 min, 30 min or overnight. Around 80% of the HCT116 cells was detected in the samples that were fixed with RBC lysis buffer for 5 min and 30 min (Figure 3C, upper row, left and middle column), while the yield decreased to 50% upon overnight fixation (Figure 3C, upper row, right column). The amount of HCT116 cells fixed with 2% PFA was around 65% (Figure 3C, lower row, left and middle column). There were no major differences between 5 min and 30 min fixation. Interestingly, the sample fixed with 2% PFA overnight completely lost EGFR expression (Figure 3C, lower row, right column), supporting sample fixation with RBC lysis buffer.

The Attune NxT is optimized for the detection of rare events in large samples. Nevertheless, we investigated whether sample filtration prior to analysis was beneficial. Healthy donor blood was spiked with HCT116 cells and analyzed either directly or after filtration using a 100 µm filter. During acquisition of unfiltered samples, in some cases, the event rate dropped entirely, causing loss of (part of) a sample (Appendix A, left panel). In others, the event rate fluctuated, resulting in irregular cell counts during analysis (Appendix A, right panel). During acquisition of filtered samples, the event rate was stable (Appendix A). Equal amounts of HCT116 cells were detected in both the unfiltrated sample and the filtrated sample, supporting that sample filtration provides a stable event rate without interfering with CTC detection.

### 2.3. Cancer Patients Have Increased Numbers of EpCAM+EGFR+ Cells in Their Circulation

Ultimately, we aim to detect CTCs in blood samples of cancer patients. For this aim, blood samples were collected from metastatic colorectal cancer patients, as well as lung cancer patients. We were able to detect EpCAM+EGFR+ events in most patient samples (Figure 4). Moreover, the amount of EpCAM+EGFR+ events in patient samples was increased compared to healthy donors, although not significantly. As healthy donors presented a relatively high amount of EpCAM+EGFR+ events, these cells were studied in more detail using the Attune CytPix (Thermo Fisher Scientific, Waltham, MA, USA). The Attune CytPix is an extension of the Attune NxT with a bright field camera, which enables researchers to investigate the morphology of a cell population identified with flow cytometry.

First, PBMCs and tumor cells were analyzed separately. As expected, PBMCs presented as small, round cells, while tumor cells showed an enlarged, more scattered morphology. When PBMCs and tumor cells were mixed, both cell populations were distinguished clearly (Appendix A). When blood samples from healthy donors were analyzed with the Attune CytPix, EpCAM+EGFR+ events showed no tumor cell-like morphology (Appendix A). This indicates that the EpCAM+EGFR+ events detected in healthy donor blood samples are false positive events (probably debris) that should be discarded. Finally, blood samples from metastatic colorectal cancer patients, as well as lung cancer patients were analyzed with the Attune CytPix. Again EpCAM+EGFR+ events were detected in most patient samples (Figure 5). However, these events showed no tumor cell-like morphology either. Most of the events were doublets, mainly consisting of either PBMCs or red blood cells, or a combination thereof. Additionally, a large proportion of events had no cellular morphology at all. This could indicate that this method is not sufficiently sensitive for the detection of low numbers of CTCs, or that these patients have no CTCs.

### 2.4. Confirmation of Tumor Cell Absence in Patient Blood Samples by CellSearch

As CellSearch is the only FDA-approved method for the detection of CTCs, this method was used to confirm the absence of CTCs in a subset of patient samples (N = 10). Therefore, blood samples from metastatic colorectal cancer patients were analyzed simultaneously, comparing the Attune NxT with CellSearch. In one out of ten patients one CTC was detected using CellSearch. However, none of the patients showed an increased aneuploidy score, indicating that the amount of ctDNA was extremely low [50]. Therefore, it was concluded that these patients had no CTCs.

## 3. Discussion

Recently, CTC detection has shown significant potential in prediction of prognosis, as well as early detection, and development and monitoring of treatment strategies in cancer patients [9,10]. Currently, CellSearch is the only FDA-approved method for the detection of CTCs [47,48]. However, as CellSearch is limited in its ability to detect CTCs in (early stages of) certain cancer types, we aimed to develop a new method for the detection of CTCs using the Attune NxT. When healthy donor blood samples were spiked with defined amounts of different tumor cell lines, most tumor cells were detected. Moreover, increased numbers of EpCAM+EGFR+ events were detected in both metastatic colorectal cancer patients and lung cancer patients compared to healthy donors. However, a relatively high number of EpCAM+EGFR+ events was detected in healthy donors as well. Although quite unlikely, this could indicate a predisposition to, or even presence of, cancer in those donors. Alternatively, classification of EpCAM+EGFR+ events as tumor cells could be unjustified. More in depth analysis of the morphology of these events proved the latter. EpCAM+EGFR+ events detected in healthy donor blood samples showed no tumor cell-like morphology. Instead, these events presented as debris that likely bound the anti-EpCAM antibodies and anti-EGFR antibodies aspecifically, causing irrelevant staining. These findings support the absence of tumor cells in the blood of healthy donors analyzed by CellSearch [48].

Current CTC detection assays, including CellSearch, are typically based on expression of epithelial cell markers, such as EpCAM. Recovery rates of CellSearch are over 85% with a clinical detection rate of more than 70% [48]. Similarly, MagSweeper is an immunomagnetic cell separator that enriches EpCAM+ target cells and eliminates cells that are not bound to magnetic particles [51]. Although clinical detection rate was 100%, sensitivity of this method was lower compared to CellSearch. However, EpCAM expression is not limited to tumor cells, as healthy epithelial cells express EpCAM as well. It has been shown that EpCAM+EGFR+ cells were found in benign inflammatory colon diseases, such as Crohn disease [52]. Events that would be classified as tumor cells were detected in a proportion of patients. Thus, although cancer patients can be distinguished from healthy donors based on the presence of EpCAM+EGFR+ cells, patients with benign inflammatory colon diseases could be misclassified as cancer patients. Thus, the definition CTC should be used with caution. This supports that there is a need for more in depth analyses of these so-called CTCs. Recently, several methods have been developed for molecular and functional characterization of CTCs [53,54]. The genotype of CTCs can be assessed by means of single-cell sequencing, including mutations originating from the primary tumor, as well as newly acquired mutations. The EPISPOT platform facilitates analysis of specific proteins secreted from CTCs [55]. Additionally, protein expression on CTCs can be studied using either Western blot or flow cytometry [56,57]. Successful culture of isolated CTCs in vitro has created the opportunity to perform functional studies as well [8,58].

Metastasis formation remains a poorly understood process. Nevertheless, it is accepted that it consists of two phases: (1) tumor cell dissemination from the primary tumor followed by intravasation into the bloodstream and (2) tumor cell extravasation into a distant organ. Tumor cell dissemination is initiated by the epithelial-to-mesenchymal transition (EMT), a process through which cells lose their epithelial traits, while acquiring mesenchymal ones, changing their phenotype. This suggests that detection of CTCs based on epithelial cell markers, such as EpCAM, may be hampered [59]. Analysis of EpCAM expression on CTCs isolated from metastatic carcinoma patients demonstrated a mean expression of around 50,000 EpCAM molecules per cell [60]. This was approximately 10-fold lower than EpCAM expression on primary and metastatic tumor tissue. Recently, it was shown that the majority of CTCs in a cohort of NSCLC patients was EpCAM- [44]. Additionally, platelets are often attracted by CTCs, forming a cluster around them [3,4]. These platelets could shield tumor cell membrane proteins, including EpCAM, complicating CTC detection. Therefore, enrichment for EpCAM+ cells, as well as staining with anti-EpCAM antibodies could result in an underestimation of the number of CTCs. To overcome these challenges, we developed a CTC detection method, which does not include an enrichment step based on EpCAM expression. Additionally, EGFR is used as a second tumor marker. EGFR is highly expressed in a variety of cancer types, including CRC [61,62,63], and treatment strategies often include anti-EGFR antibodies, such as cetuximab [64]. We anticipate that the majority of CTCs is detected using EpCAM and EGFR as tumor markers, combined with morphological confirmation. In depth analysis of EpCAM+EGFR+ events could provide valuable information on their origin, capacity to develop into metastases and response to treatment. Currently, in depth molecular analysis is not possible as the Attune NxT cannot sort cells. Possibly, this might be an option if the Attune NxT is extended with a sorting device.

Alternatively, circulating tumor endothelial cells (CTEC) have been detected in patients with a variety of cancer types. Presence of CTECs was associated with chemotherapeutic efficacy in breast cancer patients [65]. Similarly, high numbers of CTECs were correlated with resistance against immunotherapy in NSCLC patients [66]. As opposed to EpCAM and EGFR, vimentin is a mesenchymal cell marker [67]. Nevertheless, vimentin is highly expressed in multiple epithelial cancer types, and its expression has been demonstrated on CTECs. Evidence is increasing that vimentin is involved in EMT and is, therefore, associated with poor prognosis. More specifically, presence of vimentin+ CTECs, but not EpCAM+ CTCs, at baseline exhibited predictive value for poor response to treatment, and worse prognosis. [44].

Several alternative CTC detection assays that do not require an enrichment step based on EpCAM expression have been described [45,46]. The Strep-tag immunomagnetic cell separation system uses biotin-triggered decomposable immunomagnetic beads [68]. This method is equipped to include multiple antibodies simultaneously to capture a broader spectrum of CTCs. Although clinical detection rate was 100%, sensitivity of this method was lower compared to CellSearch. Methods for the identification of CTCs based on absence of certain antigens have been developed as well. Negative enrichment methods generally target CD45, which is expressed on blood cells, but not on CTCs. In this way, (all subpopulations of) CTCs can be obtained irrespective of specific antigen expression. However, these methods result in low purity, and no negative enrichment methods are currently in clinical development. Alternatively, identification of CTCs can be based on physical properties instead of antigen expression. Size-based enrichment methods depend on differences in size between CTCs and blood cells. CTCs have increased size (9–19 µm) compared to leukocytes (7–9 µm), although small CTCs might have comparable size to leukocytes. The flexible micro spring array (FMSA) was developed for the separation of CTCs based on size [69]. Additionally, density-based enrichment methods depend on differences in density between CTCs and blood cells. The most well-known density-based enrichment method is the AccuCyte system [70]. Although both FMSA and AccuCyte demonstrated increased clinical detection rates compared to CellSearch, neither method has yet advanced in clinical development.

The majority of EpCAM+EGFR+ events detected in blood samples of cancer patients were doublets of PBMCs, red blood cells or a combination thereof. However, part of these events had no cellular morphology at all. It remains unclear whether these events are dead (tumor) cells or debris. A study in prostate cancer patients demonstrated that few of the events classified as CTCs were intact cells [71]. Most of the events were either damaged cells or cellular fragments, of which some expressed apoptosis-related proteins. This suggests that many CTCs are dead upon detection.

Usually, peripheral blood samples are collected for the detection of CTCs. However, it has been shown that the number of CTCs greatly varies between blood samples [72,73]. CTC count differed in either the systemic circulation or the portal vein of pancreatic cancer patients [72]. Moreover, patients with CTCs in the portal vein had increased risk of liver metastasis development. Another study showed that 43% of lung cancer patients presented with CTCs in the pulmonary vein, while CTCs in peripheral blood were detected in only 22% of patients [73]. Again, presence of CTCs in the pulmonary vein was associated with recurrence.

Taken together, it is likely that not all CTCs are detected with the current detection methods. Additionally, detected CTCs might not be involved in metastasis formation. To address the first challenge, we aimed to develop a new method for the detection of CTCs using the Attune NxT. Although the spiking experiments demonstrated great potential, detection of CTCs in patient samples was unsuccessful. One of the possible explanations is that the detection method as described may not be sufficiently sensitive, especially in case of a small number of CTCs. Alternatively, the patients that have been analyzed could lack (detectable numbers of) CTCs. Our data combined with CellSearch data suggests the latter. Thus, analysis of a large cohort of patients with multiple tumor types and stages is required to establish the sensitivity of this CTC detection method. Therefore, the Attune CytPix method may serve as an alternative for CellSearch with respect to availability and costs, but only when a relatively high amount of EpCAM+EGFR+ CTCs is expected. Alternatively, other markers could be included to detect EpCAM- and/or EGFR- CTCs.

In conclusion, the method for detection of CTCs using the Attune NxT combined with morphological confirmation (e.g., with Attune CytPix) as described here could be used as an alternative for CellSearch. However, its use should be limited to research, and to situations in which large numbers of EpCAM+EGFR+ events are expected. The method is currently not suitable to guide clinical decisions. Moreover, it is advised that (flow cytometric) studies that either lack analyses of healthy donor blood samples or without morphological confirmation of so-called CTCs should be interpreted with great caution.

## 4. Methods

### 4.1. Cell Culture

The human epidermoid carcinoma cell line A431 and the human colorectal carcinoma cell lines HCT116 and HT29 (ATCC, Manassas, VA, USA) were cultured in DMEM (Invitrogen, part of Thermo Fisher Scientific, Waltham, MA, USA) supplemented with 10% heat-inactivated fetal bovine serum (FBS) (Biowest, Nuaillé, France), and 100 U/mL penicillin, 100 µg/mL streptomycin and 200 µM L-glutamine (Gibco, part of Thermo Fisher Scientific) (hereafter referred to as complete DMEM) under standard incubator conditions (37 °C, 5% CO_2_). The human lung carcinoma cell lines H1650 and HCC827 (provided by Jelle van der Bor, department of Medical Chemistry, Amsterdam UMC location VUmc) were cultured in RPMI (Invitrogen) supplemented with 10% FBS, 100 U/mL penicillin, 100 µg/mL streptomycin and 200 µM L-glutamine (hereafter referred to as complete RPMI) under standard incubator conditions. Cell suspensions were prepared by enzymatic digestion using trypsin-EDTA solution (Invitrogen). Viability was assessed by trypan blue exclusion and always exceeded 95%.

### 4.2. Blood Processing

#### 4.2.1. Blood Collection

Blood from colorectal cancer patients was obtained from the Biobank Hepato-Pancreato-Biliary Disease (HPB) of the Amsterdam UMC location VUmc (2018.063). Blood collection was approved by the medical ethical committee of the Amsterdam UMC location VUmc. Blood from lung cancer patients was obtained from the Liquid Biopsy Center Biobank Thoracic Oncology of the Amsterdam UMC location VUmc (2017.545) and the Fluid Phase Biopsy in NSCLC study of the Netherlands Cancer Institute—Antoni van Leeuwenhoek Hospital (NL45524.031.13). Blood collection was approved by the medical ethical committee of the Amsterdam UMC location VUmc and the Netherlands Cancer Institute, respectively. All patients signed informed consent according to Dutch and international law. As comparison, blood from healthy donors was obtained from Sanquin (Amsterdam, The Netherlands).

#### 4.2.2. Blood Processing

Whole blood (7.5 mL per sample) was diluted 1:1 in PBS and loaded on Lymphoprep (Nyegaard, Oslo, Norway), after which cells were separated by density centrifugation. PBMCs were extracted from the interphase of the Lymphoprep gradient and washed three times with PBS supplemented with autologous plasma. Samples were fixed with lysing solution (BD Biosciences, San Jose, CA, USA) and stored in 0.5% PBS-BSA. As comparison, blood from healthy donors was spiked with defined amounts of human carcinoma cells, either or not previously stained with CFSE (Invitrogen), and processed similarly. Alternatively, PBMCs were enriched for EpCAM+ cells by magnetic separation using EpCAM MicroBeads (Miltenyi Biotec, Bergisch Gladbach, Germany) before fixation.

### 4.3. Flow Cytometry

#### 4.3.1. Binding of Anti-EGFR Antibodies

Human carcinoma cells were incubated with cetuximab (Merck, Schiphol-Rijk, The Netherlands) at different concentrations for 45′ on ice. After washing, cells were stained with Alexa Fluor^®^ 488-labelled recombinant anti-human EGFR antibody (clone Hu104—R&D Systems, part of Bio-Techne, Minneapolis, MN, USA) and analyzed by flow cytometry (Attune NxT—Thermo Fisher Scientific).

#### 4.3.2. Detection of (Circulating) Tumor Cells

Samples were stained with APC-labelled mouse anti-human EpCAM antibody (clone EBA-1—BD Biosciences) and BV421-labelled mouse anti-human EGFR antibody (clone AY13—BioLegend, San Diego, CA, USA) and analyzed by flow cytometry (Attune NxT, or Attune CytPix—Thermo Fisher Scientific). CTCs were defined as EpCAM+EGFR+ cells within the live cell population based on FSC-SSC gating.

### 4.4. Statistical Analysis

Flow cytometry plots were created with FlowJo 10. Graphs were produced and statistical analysis was performed in GraphPad Prism 9. Bars depicted in the graphs represent mean ± standard error of the mean (SEM). Differences in data were analyzed with an ANOVA test followed by a Bonferroni’s multiple comparison test. *p*-values < 0.05 were considered statistically significant.

## Figures and Tables

**Figure 1 ijms-24-00021-f001:**
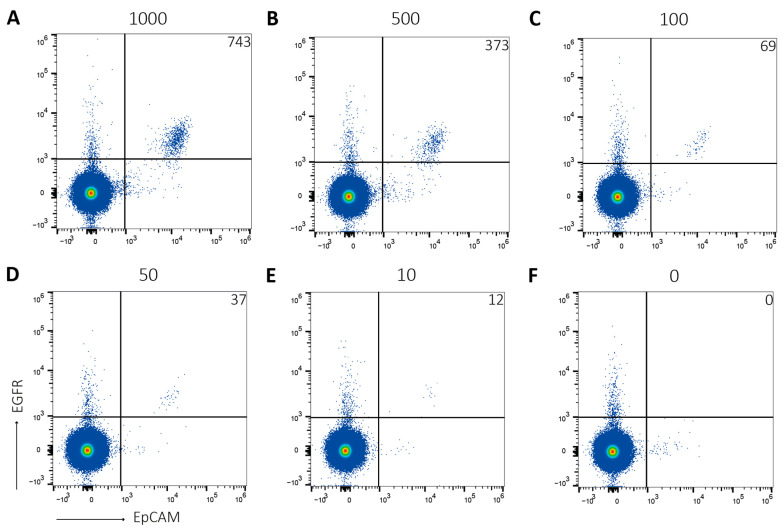
On average, 75% of tumor cells can be detected using the Attune NxT. Healthy donor blood was either (**A**) spiked with 1000 HCT116 cells, (**B**) 500 HCT116 cells, (**C**) 100 HCT116 cells, (**D**) 50 HCT116 cells or (**E**) 10 HCT116 cells, or (**F**) non-spiked. After PBMC isolation, samples were stained for EpCAM and EGFR, and analyzed using the Attune NxT. Tumor cells were detected by EpCAM+EGFR staining. Number of detected tumor cells is indicated in upper right corner. Representative figure of N = 3.

**Figure 2 ijms-24-00021-f002:**
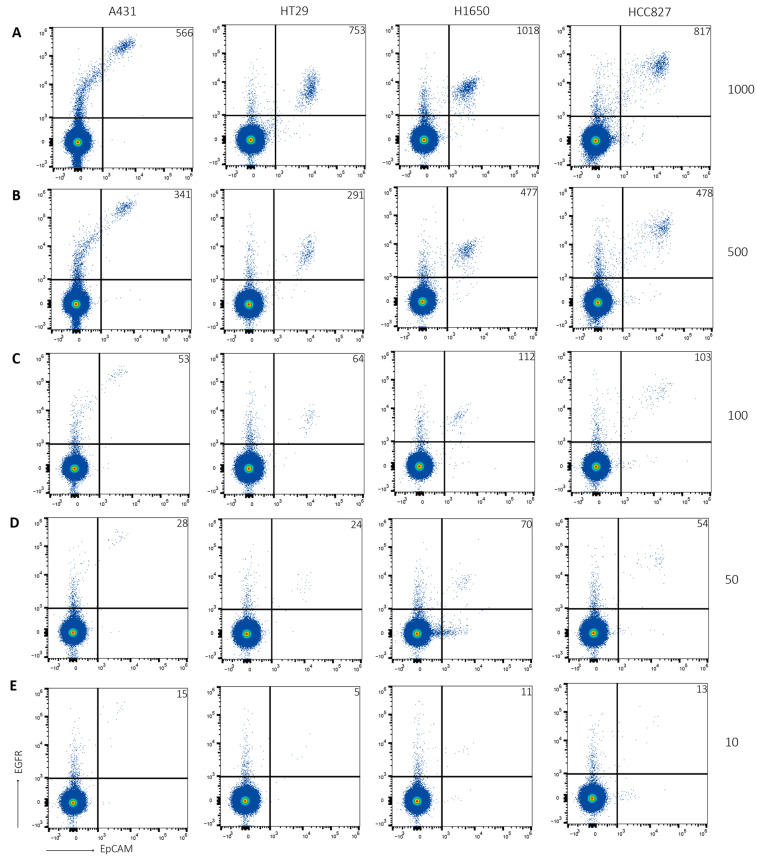
Low numbers of multiple tumor cell lines can be detected in blood. Healthy donor blood was spiked with (**A**) 1000, (**B**) 500, (**C**) 100, (**D**) 50 or (**E**) 10 A431 cells (first column), HT29 cells (second column), H1650 cells (third column) or HCC827 cells (fourth column). After PBMC isolation, samples were stained for EpCAM and EGFR, and analyzed using the Attune NxT. Tumor cells were detected by EpCAM+EGFR staining. Number of detected tumor cells is indicated in upper right corner. Representative figure of N = 3.

**Figure 3 ijms-24-00021-f003:**
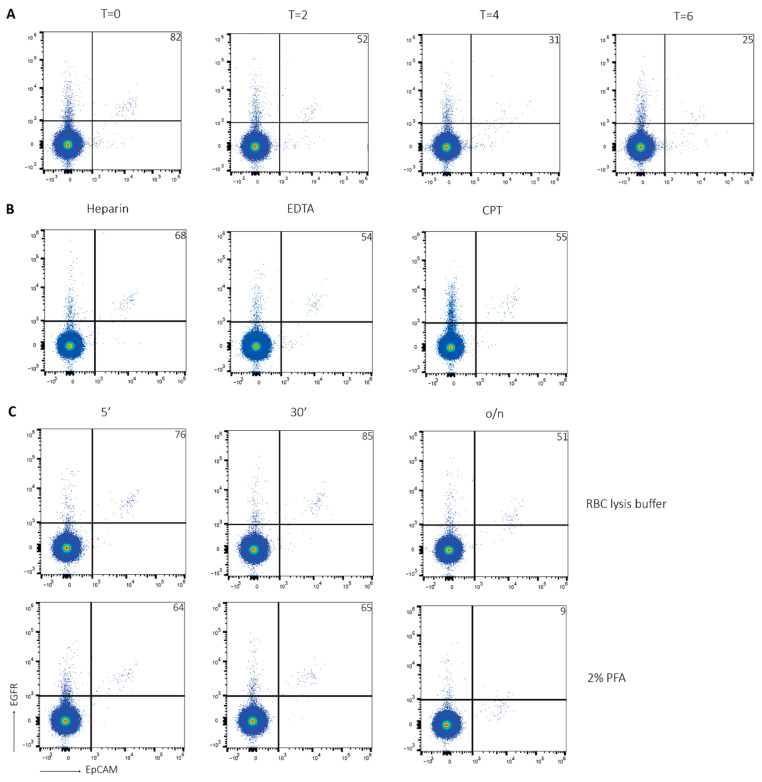
Optimization of CTC detection. (**A**) Healthy donor blood was spiked with 100 HCT116 cells and processed at T = 0 (first column), T = 2 (second column), T = 4 (third column) and T = 6 h after blood collection (fourth column). After PBMC isolation, samples were stained for EpCAM and EGFR, and analyzed using the Attune NxT. Tumor cells were detected by EpCAM+EGFR staining. (**B**) Healthy donor blood was collected in either a heparin tube (left column), an EDTA tube (middle column) or a CPT tube (right column) and spiked with 100 HCT116 cells. After PBMC isolation, samples were stained for EpCAM and EGFR, and analyzed using the Attune NxT. Tumor cells were detected by EpCAM+EGFR staining. Representative figure of N = 3. (**C**) Healthy donor blood was spiked with 100 HCT116 cells. After PBMC isolation, samples were fixated with RBC lysis buffer (upper row) or 2% PFA (lower row) for 5 min (left column), 30 min (middle column) or overnight (right column). Then, samples were stained for EpCAM and EGFR, and analyzed using the Attune NxT. Tumor cells were detected by EpCAM+EGFR staining. Number of detected tumor cells is indicated in upper right corner. Representative figure of N = 5.

**Figure 4 ijms-24-00021-f004:**
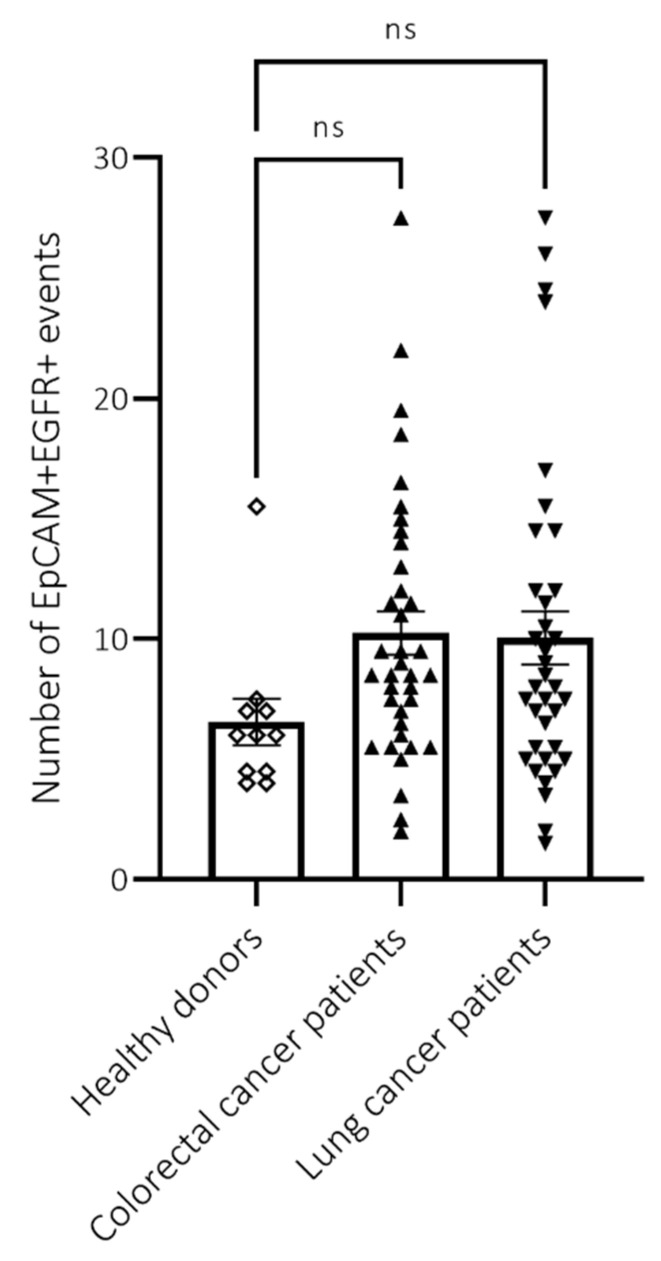
Cancer patients have increased numbers of EpCAM+EGFR+ cells in their circulation. Blood analysis of healthy donors (◊), colorectal cancer patients (
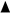
) and lung cancer patients (
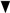
). After PBMC isolation, samples were stained for EpCAM and EGFR, and analyzed using the Attune NxT. CTCs were detected by EpCAM+EGFR staining. *p* = 0.05.

**Figure 5 ijms-24-00021-f005:**
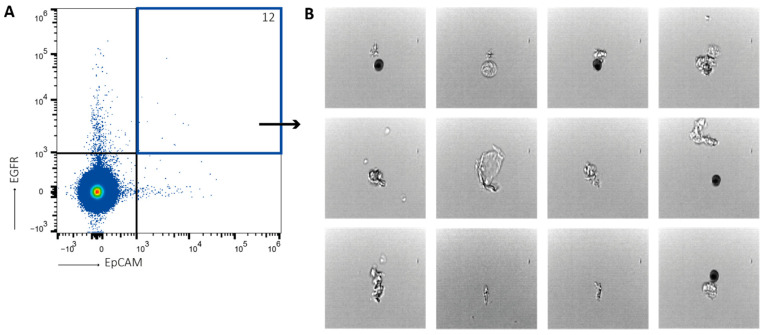
Patient EpCAM+EGFR+ cells have no tumor cell-like morphology. Blood analysis of cancer patients. After PBMC isolation, samples were stained for EpCAM and EGFR, and analyzed using the Attune CytPix. (**A**) Tumor cells were detected by EpCAM+EGFR staining. (**B**) EpCAM+EGFR+ cells have no tumor cell-like morphology. Number of detected tumor cells is indicated in upper right corner. Representative figure of N = 15.

## Data Availability

The data presented in this study is contained within the article or Appendix A.

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
