# Peer review of "Detection of Circulating Tumor Cells Using the Attune NxT"

_ijms, 2022, doi:10.3390/ijms24010021_

Round 1

Reviewer 1 Report

Low or non-expression of EpCAM    on CTCs due to EMT has been well described elsewhere. CTCs shed from primary tumor are subjected to anoikis, shear stress and immune system selection, which may result in a significant change of cell morphology. It might be too subjective to identify CTCs based upon cell morphology. Additional strategies regarding identifying CTCs, such as multiple tumor marker staining, aneuploidy of chromosomes, etc should be tested and thoroughly discussed.  

Author Response

We agree with the reviewer that cell morphology should not be the only defining factor. Hence, we use two epithelial markers (EpCAM and EGFR) to identify potential CTCs, and then use morphology as additional feature to distinguish (tumor) cells and false-positive events. Our apologies if we were unclear. We clarified the use of two tumor markers and the lack of further (genetic) analysis of CTCs in the method as described here. Moreover, we included an additional paragraph on other CTC detection methods on page 11-12.

Reviewer 2 Report

The authors of this paper set out to develop a new method for CTC detection using Attune NxT – a flow cytometry-based application specifically developed to detect rare events in biological samples without the need for enrichment. They used blood samples from healthy donors enriched with various amounts of different EpCAM+EGFR+ tumor cell lines (HCT116, A431, HT29, H1650, and HCC827), and the recovery yield was on average 75%. The detection range was between 1000 and 10 cells per sample. Cell morphology was checked with Attune CytPix. Blood samples from patients with metastatic colorectal cancer and lung cancer were analyzed, and increased EpCAM+EGFR+ events were found in more than half of the patient samples.

However, most of these cells did not show (tumor) cell morphology. Notably, the CellSearch analysis of the blood samples from a subset of colorectal cancer patients also failed to detect CTCs, suggesting that these blood samples were negative for CTCs.

The authors conclude that Attune NxT is not a better method for the detection of small amounts of CTCs than CellSearch, although sample handling and analysis are easier. Furthermore, the authors believe that the detection of CTCs using Attune NxT combined with morphological confirmation (e.g., with Attune CytPix) as described here can be used as an alternative to CellSearch. However, its application should be restricted to research and situations involving a large number of EpCAM+EGFR+. The authors believe that the method is currently not suitable for making clinical decisions. Moreover, it is advised that (flow cytometry) studies lacking the analysis of blood samples from healthy donors or without morphological confirmation of the so-called CTC should be interpreted with great caution.

Based on the results and conclusions of the author, it can be stated that it is a method with limited possibilities for clinical application. However, as it is a comprehensive study to demonstrate the possibilities of flow cytometry using the Attune NxT in the detection and quantification of CTCs, I think that it brings valuable and useful data for future research. Therefore, I recommend that the manuscript be accepted. However, it is necessary to supplement the discussion by commenting on the sensitivity of the CellSearch method and other techniques that are currently used to make clinical decisions in certain types of solid tumors. Furthermore, it is necessary to add a description of abbreviations in the article.

Author Response

We included an additional paragraph on other CTC detection methods, including their sensitivity on page 11-12. Additionally, we added a list of abbreviations.

Round 2

Reviewer 1 Report

Technic limitation of the Attune NxT described in the current manuscript is obvious, particularly in detection of low number of CTCs. CellSearch, which detects only EpCAM and Cytokeratin double positive CTCs, is apparently not an ideal gold standard. In view of the truth that majority of CTCs are EpCAM- and Vimentin- aneuploid null cells (Zhang et al., 2021 Mol Oncol 15:2891), authors are encouraged to broaden their vision to discuss recent progress  in the field.                                                                      
